# CFD Modeling of Ventilation and Dust Flow Behavior in Polishing and the Design of an Innovative Wet Dust Removal System

**DOI:** 10.3390/ijerph17166006

**Published:** 2020-08-18

**Authors:** Jianghai Qian, Junfeng Wang, Hailong Liu, Haojie Xu

**Affiliations:** 1Department of Fluid Mechanics, School of Energy and Power Engineering, Jiangsu University, Zhenjiang 212013, China; qianjianghai@sina.com; 2Department of Fluid Machinery, School of Energy and Power Engineering, Jiangsu University, Zhenjiang 212013, China; leo@ujs.edu.cn; 3Department of Engineering Thermal Physics, School of Energy and Power Engineering, Jiangsu University, Zhenjiang 212013, China; xuhaojie_ujs@126.com

**Keywords:** polishing, aluminum dust, CFD modeling, wet dust collection, multiphase flow transport

## Abstract

Fine aluminum dust pollution in the polishing process was detected during a field survey. To obtain a fundamental understanding of the airflow patterns and the fine dust dispersion characteristics during a polishing process, computational fluid dynamics simulations were first performed to analyze the data collected in field measurements. The inappropriate ventilation arrangement and lack of effective dust control measures were identified as the main reasons for the high dust exposure levels (in excess of 1000 μg/m^3^). Simulation results showed that inhalable dust particles (PM10) could be significantly diluted at the operator’s breathing level by adding a supply air inlet above the operating area. Moreover, dry dust collection systems create a risk of aluminum dust explosion accidents. An innovative design of wet dust removal system which could mitigate the occurrence of dust explosions was proposed and then implemented on site. An independent field dust assessment showed that a reduction of fine dust particles up to 95% in the worker’s breathing area and the fine dust in the vents was reduced to 80%. Therefore, the proposed strategies are implemented immediately to address the combustible dust in the polishing working environment and can provide guidance for operators.

## 1. Introduction

With the rapid development of the aluminum products industry, aluminum and aluminum alloys are one of the most widely used nonferrous structural materials in industrial applications which have been widely used in the aerospace, automobile, mechanical manufacturing, and chemical industries [1]. As a result of the metal surface requirements, raw metal must be polished. However, the polishing process produces a lot of metal dust which introduces occupational hazards to workers, such as aluminosis (also known as “aluminum lung”), significantly affecting the health of employees. Furthermore, dust emissions into the atmosphere causes environmental pollution. More than 975,000 cases of occupational illness occurred in 2018, and 90% of reported occupational diseases were pneumoconiosis in China [2]. In recent years, new cases of worker pneumoconiosis have risen continuously every year. According to data from the Chinese Center for Disease Control and Prevention, an average of 20,000 to 250,000 new cases of pneumoconiosis have been reported annually since 2010, and the actual number of cases was much higher than the number of reported cases (CDC, 2019) [3].

Aluminum powder is a kind of combustible dust which is a serious threat to safety production [4]. When the concentration of aluminum dust dispersed in the air reaches a certain explosion limit and encounters an ignition source, severe dust explosion accidents can occur [5]. Eckhooff noted that accidental dust explosions remain a constant danger in processing industries [6]. In addition, Taveau et al., reviewed seven accidents that occurred in aluminum processing facilities [7].

The amount of dust that was reluctantly generated during polishing processes should be removed and collected in a proper collection system. Cyclone and bag filter were the most commonly used reduction systems in polishing process. However, dry dust collection systems create a risk of aluminum dust explosion accidents. According to Taveau et al. (2018), about 65% of the explosions that involved metal dust occurred in dust extraction and abatement systems in Germany in the 1965–1980 period. As outlined by Taveau et al. (2015), fine particles generated from production become blocked in filters, and their concentration in the air can exceed the minimum explosible concentration (MEC) level. On the morning of 2 August 2014, a catastrophic explosion of aluminum alloy occurred in a polishing workshop in Kunshan, China. The explosion killed 146 people, and the direct economic loss was 351 million Yuan [8]. This accident was a serious waring of the dangers of industrial dust explosions. Therefore, there was an urgent need for new knowledge and awareness to prevent such future disasters.

Domestic researchers have been working on various dust control measures. Based on a polishing machine’s processing features and dust characteristics, Yu et al. (2013) performed computational fluid dynamics (CFD) simulations and optimized the structural design using the results. Suction hoods were designed based on experience and tests of the flow field smoothness [9]. Chen studied the movement and deposition of aluminum dust in a dust collection pipeline, found the most suitable airflow velocity for the dust deposition interval and transport, and provided guidance for the optimization of the wind speed of the dust removal system [10]. The influencing factors of the aluminum dust deposition in a dust collection pipeline were examined to avoid the explosion concentration. Hua et al., analyzed a ventilation dust removal system and found that there are many potential safety hazards in the design, operation, and maintenance processes [11]. All these studies indicated that dry dust collection systems usually use ventilated dust removal technology, which has the disadvantages of complicated dust collection pipelines, easy formation of blockages, high energy consumption, and large potential safety hazards. By contrast, wet dust removal system can effectively reduce the risk of dust explosions.

Spray dust reduction is widely used in dust pollution control. As early as 1976, Brown reported that fine water mist was effective in removing respirable dust [12]. Liu et al. (2001), by using a screen preliminary disposal system and liquid-film sprinkling techniques, removed the dust of automobile tire keel burnishing workrooms. This technology reduced the dust emissions to some extent, but the final emission concentration of the dust exceeded the values stipulated in the latest laws and standard requirements [13]. Stulov studied the collision and capture efficiency of solid aerosol particles from a water surface experimentally [14]. By combining orthogonal design and experimental results, Wang et al., derived swirl-pressure nozzles, which can provide guidance and suggestions for establishing a dust suppression scheme using spraying [15]. The field test showed that the control of respirable dust by ultra-fine water droplets were effective, and the reduction efficiency could reach up to 35% [16]. In order to reduce dust effectively, the fine water mist system needed to be further optimized. Trials with a water mist spraying system also showed promising results with a 40% respirable dust mitigation effect [17]. However, there have been few studies on wet dust removal in polishing industry.

More recently, Mohamed et al. (2019) simulated the dispersion of ultrafine particle during rotary polishing. The results showed that the suction flow rate and the speed of rotating disk have obvious effects on the performance of local exhaust ventilation system. which illustrated the complexity of the dust removal in polishing processes [18]. Saidi et al., studied dust particle dispersion and the surface quality during dry polishing of granite [19]. Jiang established a dust distribution model of mobile grinding operations and investigated the influences of the dust source strength, average particle size, wind speed, and atmospheric stability on the dust distribution [20]. These studies improved researchers’ understanding of the ventilation and dust particle flow dynamics under specific polishing conditions, which was of great significance for developing polishing dust control measures/equipment.

In addition, governments have enacted mandatory measures to provide cleaner and safer working environments for workers. China formulated relevant dust emission standards and dust removal system safety standards, including GB 15577-2007 “Dust explosion-proof safety regulations” and GB 17269-2003 “Aluminum and magnesium powder processing dust explosion-proof safety regulations.” Ambient air quality standard in China has been further lowered from 500 to 300 μg/m^3^, and the fine particulate matter (PM2.5) concentration limit was increased (GB/T 3095-2012 standard, 2016) [21]. Work safety and air pollution control have become China’s priorities which also promote the innovation and development of dust removal technology. However, dust control during polishing operations highly depends on many different working points, the operation being performed, and the operating parameters. Thus, a better understanding of airflow pattern and the dispersal behavior of fine particles will help to identify the causes of high dust concentration exceeding their limits and workers managed combustible dust more effectively.

In recent years, many numerical simulations on the dispersion behaviors of dust have been conducted in order to solve occupational health hazards. Torano combined field tests and numerical simulations to study the migration behavior of dust in air [22]. Balusu et al., modeled the ventilation and respirable dust flow characteristics around a shearer and evaluated the effects of different dust control measures [23]. A ventilation system with multiple outlet nozzles was simulated by Kurnia et al. The results showed that the ventilation system can not only meet the requirements of ventilation, but also save energy significantly [24]. Ali Bahloul et al., modeled and measured dispersion and distribution of ultra-fine particles (UFP) in a granite polishing process [25]. The results will be helpful for further understanding the influence of the use of ventilation on the airflow/dust dispersion behaviors. This could provide more appropriate guidance for the development and design improvement of dust removal technology.

It is noted from these studies that the safe treatment of aluminum dust not only required adequate protection methods but also needed to improve the risk awareness of employees and a safe dust removal system. The dust movement in the airflow field in polishing and wet dust removal technology has been insufficiently investigated. Therefore, the ventilation and dust flow behaviors were studied by multiphase flow simulation in a polishing process.

The problems of an existing system were identified. Finally, a new dust mitigation system that uses wet dust collection was proposed, its performance on dust control and reduction was evaluated, and it was implemented in the field. The proposed measurement can overcome the shortcomings of traditional dust removal equipment for suppressing the dust produced during polishing. By doing so, on the one hand, the dirty airflow could be drawn into the dust collection channel. On the other hand, the wet dust collection system could suppress the dust effectively and safely.

## 2. Site Condition and Ventilation Survey

The main purpose of the polishing survey and ventilation survey were to understand the problems existing in the existing system and to gather basic information e.g., the dimensions of the polishing machine, the layout of the polishing and ventilation systems, and existing dust control measures.

### 2.1. Site Conditions

Figure 1 shows a view of the polishing chamber. Field investigations were conducted in the operator area where the polishing machine was polishing. Originally, the polishing machine was equipped with a dust collector channel (Figure 1). However, its dust removal performance was very poor. The designed cross section of the system had a 2.75 m width and 2.3 m height. The polishing chamber had an open inlet, a strip inlet at the bottom of the dust collecting channel, and an exhaust fan at the outlet of the dust collecting channel. Preliminary analysis showed that this ventilation arrangement may be the main reason for the failure of the dust collector.

During the survey, it was observed the dust control was not conducive to the polishing process. First, the polishing operation area was not completely closed, resulting in an insufficient suction capacity of the exhaust fan. This caused the dispersal of dust particles into the worker area and the environment, and therefore, existing system needed to be designed and improved. Second, due to insufficient maintenance of equipment, the baffles and air filters on the dust collection channel were blocked, and the resistance was excessive, leading to flow resistance increases, wind speed decreases, and large potential safety hazards. Therefore, the absence of effective dust control measures resulted in serious dust pollution and explosion.

### 2.2. Ventilation Survey

Numerical simulations required a set of field measurements to be calibrated and validated. In order to develop an effective dust suppression strategy during a polishing process, the ventilation and dust flow characteristics must be understood first within the environment. Therefore, ventilation measurements were made during the polishing to obtain the necessary data.

Figure 2 showed the master plan for the polished dust collecting room and ventilation arrangement. The suction fan was installed at the exit of the dust collection channel, so the polished dust collecting room formed a negative pressure ventilation system, which provided power for the transport of multiphase flow. In order to investigate of ventilation performance, we used Kestrel handheld anemometer to first measure airflow at three points. (A, B, and C, as indicated in Figure 3a). According to the airflow velocity measured at points in Table 1, the gas flow into the polishing chamber about 1.0 ~ 1.2 m^3^/s.

In addition, the airflow velocities at several points in the polishing chamber (shown in Figure 3a) were measured using a Kestrel handheld anemometer, and the results are provided in Table 1. These data were used for model validation.

During the polishing process, there was mainly due to occupational health hazards and safety hazards by PM10 suspended in polishing. Consequently, we used DT-96 particle detector which used laser sensor to identify the fine particulate matter in the air to accurately measure the mass concentration of PM10 particles in real time. We measured data at points for 10 s in real time. Through statistical analysis, the statistic average value was used. In order to investigate of dust capture performance, and verify the accuracy of the numerical simulation, the measurement results of PM10 concentration at the measuring points were shown in Table 1.

## 3. Airflow Pattern and Dust Flow Modeling

### 3.1. Mathematical Models

The airflow in the polishing process followed the continuous medium hypothesis and the law of conservation of mass, momentum. For the calculation of turbulent flows, the turbulence closure problem was solved using the standard k−ε model. The dust particles produced in polishing were dispersed in the airflow. Therefore, the commercial CFD code ANSYS Fluent, which is a numerical simulation software based on the Euler–Lagrange equation. Specifically, the gas-phase airflow under the Euler coordinate system were treated as a continuous medium, the solid-phase dust particles under the Lagrange coordinate system were treated as discrete system. Moreover, the interaction between the gas-phase airflow and discrete-phase dust particles were considered.

#### 3.1.1. Mathematical Model of Air Flow

As the dust flow behavior relies heavily on the fluid flow, it is essential to obtain a flow field capable of representing the actual site conditions. The continuity equation can be express as follows:(1)$$\frac{\partial \rho }{\partial t}+\frac{\partial }{\partial x_{i}}\left(\rho u_{i}\right)=0$$

The Navier–Stokes equations can be expressed as follows:(2)$$\frac{\partial }{\partial x_{j}}\left(\rho u_{i}u_{j}\right)=-\frac{\partial p}{\partial x_{i}}+\rho g_{i}+\frac{\partial }{\partial x_{i}}\left[\left(\mu +\mu_{t}\right)\left(\frac{\partial u_{i}}{\partial x_{j}}+\frac{\partial u_{j}}{\partial x_{i}}\right)\right]$$
where ρ denotes the fluid density, $x_{i}$ and $\;x_{j}$ denote the coordinates in the x- and y-directions, respectively, $u_{i}$ and $\;u_{j}$ denote the time-averaged velocity in the x- and y-directions, respectively, p denotes the effective turbulent pressure, $g_{i}\;$ denotes the gravitational acceleration in direction *i*, and $\mu_{t}$ denotes the coefficient of turbulent viscosity.

The finite volume method, implemented in Fluent, was used to convert Equations (1) and (2) into a discretized algebraic equation form. Thus, by using iterative numerical algorithms, the physical variables of the airflow field at discrete points or volumes could be determined.

The airflow in a polishing process is a typical turbulent flow. In this study, the two-equation turbulence model (Launder and Spalding) [26] was adopted to simulate the turbulence. For the three-dimensional incompressible fluid flow, the model is given as follows:

Turbulent kinetic energy equation:(3)$$\frac{\partial }{\partial t}\left(\rho k\right)+\nabla \cdot \left(\rho k\vec{U}\right)=\nabla \cdot \left[\left(\mu +\frac{\mu_{t}}{\sigma_{k}}\right)\nabla k\right]+G_{k}-\rho \varepsilon$$

Turbulent dissipation rate equation:(4)$$\frac{\partial }{\partial t}\left(\rho \varepsilon \right)+\nabla \cdot \left(\rho \varepsilon \vec{U}\right)=\nabla \cdot \left[\left(\mu +\frac{\mu_{t}}{\sigma_{\varepsilon }}\right)\nabla k\right]+C_{1\varepsilon }\frac{\varepsilon G_{k}}{k}+C_{2\varepsilon }\rho \frac{\varepsilon^{2}}{k}$$
where k is the turbulent kinetic energy (m^2^/s^2^), ε is the turbulent dissipation rate (m^2^/s^3^), $G_{k}$ represents the generation of turbulent kinetic energy due to the mean velocity gradient, $C_{1\varepsilon }$ and $C_{2\varepsilon }$ are model constants, $\sigma_{k}$ and $\sigma_{\varepsilon }$ are turbulent Prandtl numbers for k and ε, respectively, μ is the dynamic viscosity coefficient (Pa s), and $\mu_{t}$ is the turbulent viscosity coefficient given by Launder and Spalding as follows:(5)$$\mu_{t}=\rho C_{\mu }\frac{k^{2}}{\varepsilon }$$

Based on the values recommended by Launder et al., which were later proven by experiments, the values of the model constants were $C_{1\varepsilon }=1.44$, $C_{2\varepsilon }=1.92$, $\sigma_{k}=1.0$, $\sigma_{\varepsilon }=1.3$, and $C_{\mu }=0.09$.

#### 3.1.2. Modeling of Fine Dust Flow Behavior

Based on the predicted airflow field, the aerodynamics of respirable particulates could be investigated usingComputational Fluid Dynamics(CFD) coupled with-Discrete Phase Model(DPM), which can adequately describe the migration behaviors of dust particles, was used in the present study to calculate the motion trajectory of the discrete phase dust particles, as well as the mass and momentum exchange among the airflows [27].

By analyzing the balance of forces on the particles, the motion equation can be written as:(6)$$\frac{d\vec{u_{p}}}{dt}=F_{D}\left(\vec{u}-\vec{u_{p}}\right)+\frac{\vec{g}\left(\rho_{p}-\rho \right)}{\rho_{p}}+\vec{F}$$
where $\vec{u_{p}}$ denotes the particle velocity vector, $\vec{u}$ denotes the continuous-phase velocity; $\vec{F}$ denotes additional forces,; *ρ* and *ρ_p_* denote the fluid and particle density, respectively, and $\vec{g}$ is the gravitational acceleration.

Assuming the aluminum dust particles to be spherical [2], the drag force can be expressed as follows:(7)$$F_{drag}=F_{D}\left(\vec{u}-{\vec{u}}_{p}\right)=\frac{18\mu }{\rho_{p}d_{p}^{2}}\frac{C_{D}R_{e}}{24}\left(\vec{u}-{\vec{u}}_{p}\right)$$
where $R_{e}$ denotes the relative Reynolds number, defined as $R_{e}=\rho d_{p}\left|{\vec{u}}_{p}-\vec{u}\right|/\mu$.$\;C_{D}$ denotes the drag coefficient. $C_{D}$ can be calculated as follows: $C_{D}=a_{1}+a_{2}/R_{e}+a_{3}/R_{e}^{2}$, where $a_{1}$, $a_{2}$, and $a_{3}$ are constants.

### 3.2. Geometrical Model

According to the measurement of the geometric parameters of the dust collecting system in the field polishing, the geometric model was built in 1:1 size scale by Solidworks, and is given in Figure 3a. The polishing machine is located in the middle of the operating area. Dust collection channel is located at the rear of the polishing machine. The slit suction outlet connects the dust collection channel and the operating area. The baffle is located inside the dust collection channel, at a distance of 0.7 m from the bottom. The main geometric features of the CFD model is shown in Table 2.

The geometric model is imported into the Integrated Computer Engineering and Maufacturing(ICEM)-CFD for meshing division. To avoid any mesh dependency, meshes with different densities, denoted as coarse mesh, middle mesh and fine mesh, whose total mesh numbers were about 1.8 million, 2.4 million and 3.6 million, respectively. A straight line at 1.5 m above the floor (as shown in Figure 3a) was set in the mode. Twenty points were arranged above the line at the same intervals, and the dust concentrations at these points in the three types of meshes were compared. Comparison showed that 2.4 million elements could achieve independent solution. Therefore, middle mesh was selected in this study for further calculations. The final meshing result is shown in Figure 3.

### 3.3. Computational Conditions

#### 3.3.1. Particle Size Distribution of Aluminum Dust of the Dust Sources

The source of dust in this study was from the surfaces of the polishing pieces. Particle image analyzer was adopted to analyze the sizes of dust sample, which were collected from dust sources. The analysis indicated that dust with a size smaller than 20 μm generated in polishing. According to the regression analysis of least squared method, the sizes of fine dust followed the Rosin–Rammler distribution. In addition, based on the field measured PM10 dust concentrations and particle mass fraction, dust concentration with particle size larger than 10 μm was calculated. Dust particles (sizes varied between 1–20 μm [2]) were then ‘released’ from the source in the simulation. Setting of parameters of dust source was shown in Table 3.

#### 3.3.2. Boundary Conditions and Computational Method

Boundary conditions are the key to accurately predict physical phenomena. Boundary conditions of numerical simulation are set as shown in Table 4.

## 4. Base Models Validation and Results

### 4.1. Base Models Validation

In addition to the mesh independent study, the base model results must be validated against data from field measurements before accepting the model for parametric studies. In this study, the field-measured airflow velocities and PM10 dust concentrations during the ventilation survey were employed for base model validation. Table 5 showed the comparison between the model-estimated and field-measured air flow velocities and dust concentrations. It is found that the velocity error is large at position 1#, because the velocity measured in the field is affected by the flow disturbance, resulting in a large calculation error. The predicted values are basically consistent with measured values of the remaining measurement points. Therefore, this model could be used to further investigate the airflow patterns and dust concentration in polishing process.

### 4.2. Simulation Results of Airflow

The study of air flow is helpful to develop effective dust suppression strategies. Field investigation and simulation results showed that the airflow in the operating area was greatly affected by suction and surrounding environment. The operator would stand in front of the polishing wheel and breathed at a height of about 1.5 m above the ground. Thus, simulation analysis was primarily analyzed at this level.

Figure 4 depicts the velocity contours in front of the polishing wheel. The airflow velocity showed a basically symmetric distribution in the polishing chamber. In the side area of the polishing machine, the airflow velocity increased in a gradient from top to bottom. Although air inlet speed in the operating area was higher than that in the non-operating area, there was still a lot of air following into the equipment directly from the non-operating area, and this part of the air did not play a role in carrying pollutants away. Moreover, the airflow velocity in the operating area was only 0.16 m/s, and thus, it was vulnerable to the influence of the external airflow, which could lead to particle dispersion. The polishing-wheel-induced airflow resulted in greater dispersion of dust particles in the work area, which weakened the ventilation dilution effect especially in the absence of an effective dust reduction strategy. Thus, many fine particles were suspended in the worker’s breathing area.

Figure 5 illustrates the velocity vector distribution across the polishing wheel center. Air passed through the zones of the polishing machine and was drawn into the bottom of the dust collection channel by suction, after which it was exhausted. From the strip outlet at the bottom to the open inlet, the wind speed decreased. In particular, when the inlet air arrived at the polishing pieces, the airflow became disordered and formed a series of vortices, and the particle flow trajectories were affected mainly by these vortices. These vortices also made dust particles hard to settle. The flow patterns were complex in the dust collection channel, where flow circulation under the baffle was evident. The existence of flow circulation or vortices was not favorable for the dilution of dust. Many dust particles became trapped in the circulation area. This led to significant dust contamination around the polishing machine and in the dust collector channel.

### 4.3. Simulation of Dust Flow Behavior and Distribution Patterns

Significant dust is generated during polishing process. Investigating the dispersion of dust particles will help design solutions to capture dust in polishing process. During the polishing process, dust is mainly generated by polishing the piece surface. Therefore, respirable dust particles are released from the source. The flow behavior of dust particles was investigated using the particle trajectories and dust concentrations.

Figure 6 shows the dust particle trajectories in the polishing chamber. The suction capacity of the exhaust fan was limited, dust generated from the face of the part being polished diffused quickly and dispersed widely in the entire working environment. The measurement of dust concentration of PM10 measurement in different points in Table 2 can explain the characteristics particle trajectory. During dust dispersion, dust separation occurred. Dust particles with larger diameters were more likely to settle under the action of inertial force. Dust particles with smaller diameters tended to become suspended and travel with the airstream. Meanwhile, a significant number of dust particles motion around polishing machine, causing significant contamination where operators conduct their work. Additionally, a certain number of dust particles was deposited on the floor. Thus, it is necessary to maintain a relatively high moisture content on the floor to avoid the re-entrainment of dust particles to the air. Many particles were drawn into the dust collection channel by suction, forming a high concentration of dust, which may pose a severe explosion hazard. Therefore, it is urgent to design safe and effective dust suppression measures.

It is worth noting that the calculated particle trajectories shown in Figure 6 could only be used to describe the distribution of respirable dust within the polishing process qualitatively rather than quantitatively, which is of more concern to engineers or operators. The mass concentration of respirable dust was therefore employed to quantitatively evaluate the dust pollution severity in the polishing environment.

Figure 7 depicts the dust concentration distribution 1.5 m above the floor. Due to original system poor dust capture performance, dust inevitably would flow toward the worker around the polishing machine. The dust concentration was more than 100 ug/m^3^ at the worker’s breath level. Therefore, it is recommended that the operator wear a mask and take other safeguard procedures to minimize the exposure chance. Otherwise, measures to repel dust must be taken with increasing ventilation to discharge and dilute the gas in worker operating areas. In this study, an air supply system fan filter unit (FFU) was used to supply fresh air and effectively control dust dispersion to the worker and environment.

Figure 8 depicts dust concentration across the polishing wheel center. During polishing process, the dust concentration distribution was non-uniform. The dust concentration in the dust collection channel was higher than 1000 μg/m^3^. The velocity vectors shown in Figure 6 could clearly illustrate the distribution law of dust. Figure 6 shows that as the air entered the operating area from outside to the air suction inlet, it blew most of the dust to the dust collection channel, causing significant dust accumulation, which was relatively high near the baffle. The airflow generated vortices at the baffle, and fine particles followed the airflow and gathered at the baffle. Dust particles tended to accumulate in the zones under the baffle. Dust concentration gradually dropped in the air filter and decreased further in the exit flow. However, the average concentration remained high and varied between 400 and 600 μg/m^3^. Therefore, spray system nozzles installed at the bottom of the dust collection channel, which formed a mist covered the dust sources and encouraged dust settling.

## 5. Design of New Ventilation and Wet Dust Removal System

Based on the analysis of simulation results, improved original ventilation system and a new wet treatment system was used to improve the effective control of dust in polishing process. The improvement measures were as follows:(1)The intake air supply system with FFU was moved to the top side of the operator area.(2)Sprayers were installed at the bottom of the dust collection channel, and a pool was set at the bottom of the polishing chamber. A water film plate was arranged above the inlet strip suction, as indicated in Figure 9

To understand how the new system operated, simulations and experiments were conducted using the new design. We designed a push-pull ventilation system which consisted of supply ventilation with FFU and suction ventilation with exhaust centrifugal fan. FFU was fixed above the operator to provide fresh air and blown air to push the particles ejected by the rotating disc to the exhaust. While FFU work with the exhaust centrifugal fan. It would then be easier to draw the dust laden air into the dust collection channel with the aid of the pull flow. The designed wet dust removal system would be effective for preventing aluminum dust explosions and reducing the dust concentration.

The proposed ventilation and dust removal system was analyzed in detail to check airflow and dust flow behavior. The design of the wet dust removal system is described as below subsections.

### 5.1. Simulation of Air Flow

Figure 10 shows the velocity contours in front of the polishing wheel. As a result of the new ventilation system, air flow patterns changed significantly. Compared with the original flow field shown in Figure 4, by adding an FFU at the upper side of the operator area, an effective air supply and suction system was formed. In the side area of the polishing machine, the airflow velocity decreased from the top to the bottom. Moreover, air velocity in the operating area was improved to 0.5 m/s, which effectively controlled the dust transport, preventing dusty air from dispersing into working area and environment.

Figure 11 shows the velocity vector distribution across polishing wheel center. Due to the ventilation method of the upper and lower rows, fresh air flowed from the top to the bottom, which could effectively prevent the spread of dust in the horizontal direction, and the fresh airflow preferentially passed through the breathing zone of the workers, while the majority of the dust laden air from the polishing piece surface flowed toward the plate above the strip suction outlet and the floor. The dust barely diffused by the induced and disturbed airflow around the polishing machine, and it was then drawn into the dust collection channel by the exhaust fan. The airflow velocity in the dust collection channel increased. When the airflow met the baffle, the cyclone strength increased, which was not only beneficial to the interactions between droplets and dust particles but also to the centrifugal separation of the particles.

In conclusion, the new push-pull ventilation system could form an effective negative pressure and blow fresh air through the worker breathing area, which not only prevented dust from diffusing around the polishing machine and outward but also controlled the dust transport to the dust collecting channel.

### 5.2. Simulation of Dust Flow Behavior and Distribution Patterns

The same boundary conditions as shown in Table 3 were used to set the particle source parameters to study the effect of the new ventilation and dust mitigation system. The dust particles generated during polishing dispersed characteristics were shown in Figure 12.

The new system could effectively control dust particle migration and restrained dust dispersion to workers’ breathing area. The new ventilation system helped to transport most dust particles to the dust collection channel, resulting in more ejected particles being effectively captured by the liquid level and settling quickly. Most of the dust particles were dropped into the tank on the floor. Generally, the particle trajectories indicated that the dust control effect was good.

The distribution of dust mass concentration could more quantitatively indicate the performance of the new system. Dust concentration distribution 1.5 m above the floor was shown in Figure 13. According to the concentration distribution of the results of the current model shown in Figure 7, the maximum dust concentration was set at 100 μg/m^3^. Compared with the original system, the new system has obvious dust mitigation effect. The dust concentration was below 50 μg/m^3^ in workers’ breathing area. The dust concentration was relatively lower around the workers, which could provide a healthier working environment for the operator. The results showed that the system can collect more than 95% of fine dust particles.

The dust concentration distribution across the center of the polishing wheel is illustrated in Figure 14. Dust generated from the polishing face was forced to flow toward the exhaust side. Compared with the basic model shown in Figure 9, the same maximum dust concentration was set at 1000 μg/m^3^ in Figure 14a. The results showed that the dust concentration in the collection channel decreased further with new system. However, when the maximum dust concentration was set at 500 μg/m^3^ in Figure 14b, the dust concentration was higher (above 500 μg/m^3^) under the baffle. This was mainly caused by the poor performance of the dust collection channel. Therefore, this analysis can be used to determine the optimal arrangement of the spray dust-settling equipment on site, which can greatly improve the effect of spray dust removal.

### 5.3. Field Implementation of Novel Wet Dust Collection System and Field Demonstration

A new wet dust removal method was added to suppress the dust in the polishing chamber. As shown in Figure 14b, dust concentration was high in the zones at the bottom of the chamber and the dust collection channel. Thus, a water tank was arranged at the bottom of the chamber, and a water film plate was set above the suction inlet (as shown in Figure 9). When dust laden air passed through the liquid surface, many particles collided with the liquid membrane and were captured by the moisture. Finally, the wastewater dropped into the water tank under gravity. The dust collection channel also provided a passage for the dust to flow into the exhaust. Therefore, sprayers were installed at the bottom of the dust collection channel to help capture some of the dust particles. A significant quantity of droplets was retained to disperse in the confined zones under the baffle. A swirling flow formed at the baffle, which could not only extend the residence time of the particulate matter but also strengthen the interaction between the particles and spray, and it facilitated the absorption and removal of dust. This helped to minimize the escape of dust particles at the exit. Generally, it was inferred from the profiles that by using a water mist, the majority of the contaminated ventilation flowed toward the dust collection channel inlet, through which the majority of fine dust particles were captured.

An investigation line 1.5 m above the floor and various measurement points (as shown in Figure 3) in the chamber were employed to quantify the dust concentration reduction in the polishing. Figure 15 showed the variation of dust concentration along the survey line before and after application of integrated dust control system. The dust suppression effect of the new ventilation system was obvious. The dust concentration along the line could be reduced significantly to about 20 μg/m^3^, indicating that a dust reduction efficiency of more than 95% could be achieved. With the increase of wet dust removal system, the overall dust concentration dropped gradually. The average dust concentration was generally lower than 250 μg/m^3^ in the dust collection channel. By installing the innovative wet dust suppression system, the improvements in the air quality achieved significantly improved the quality of fresh air to be delivered to workers and increased the polishing environment safety.

## 6. Conclusions

The characteristics of air flow and dust distribution were basically understood by CFD modeling method in polishing process. The dust dispersion behavior was investigated by visualizing the trajectories of individual particles and the mass concentration distributions in the integrated system. The simulation results showed agree well with that by measured in field practice.

Based on the existing dust collection system analysis, the main problems were insufficient capacity and low collection efficiency. A new dust control strategy was proposed to replace the existing dust collection system. First, the air inlet was installed directly above the operator and appropriately increased the airflow volume of the outlet suction fan, push-pull airflow from the operator side to the dust collection channel side formed. Thus, the majority of the dust laden air could flow into the dust collection channel, which was favorable for the capture of dust particles. Furthermore, use of a clean air intake (FFU) could control the dust more effectively, especially fine dust. This could effectively reduce the dust concentration around the worker breathing area and provide clean air to the worker operating area. To improve the working environment during polishing, push–pull ventilation is an effective method to control dust pollution.

Second, wet dust removal technology was implemented for the suppression of dust produced during polishing. It was found that the bottom area of the dust collecting channel was the main channel for dust diffusion to the exhaust outlet. To efficiently capture the dust, spray and liquid membranes were proposed to absorb fine particles and purify dusty air. Particles were captured by moisture and were then directly deposited by gravity into the flume. Small dust particles could easily enter the duct via the airflow and be removed by the water mist. Adding fine water mist could effectively reduce the dust concentration in the dust collecting channel.

The dust reduction effect of the new system compared with the original dust control measures was studied. Push-pull ventilation and dust removal technology could increase the dust removal rate in the breathing area by 95% on average, and the dust mass concentration was no more than 30 μg/m^3^ Wet dust removal systems not only exhibited fine dust mitigation efficiency of 80% but also effectively prevented aluminum dust explosions during metal polishing.

From this study, the ventilation arrangement and dust control measures are typical technical matters that can be optimized by conducting modeling studies and field tests. Through the effective integration of technology and management, it was concluded that a cleaner and safer working environment could be created. The results revealed from this study could be referred to by engineers working under similar production conditions. Future work should be conducted that considers the impact of the polishing wheel rotation motion on the dust control and hydrogen explosion control measures for aluminum wet dust removal systems.

## Figures and Tables

**Figure 1 ijerph-17-06006-f001:**
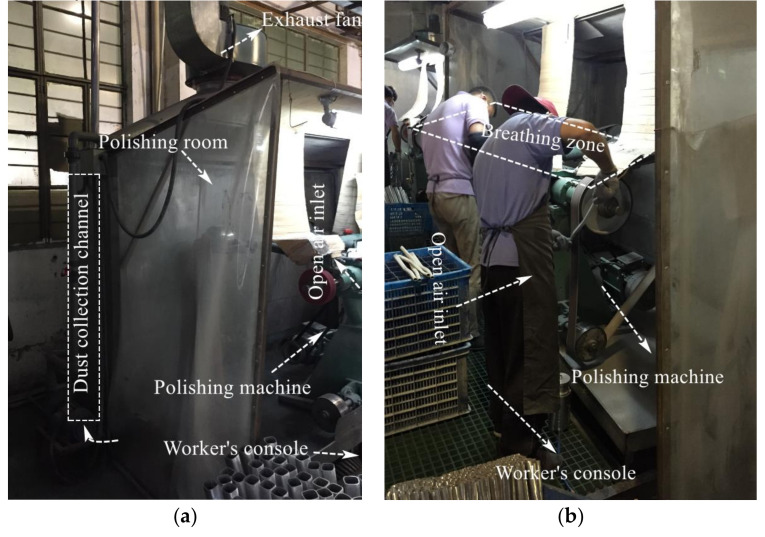
Polishing process and ventilation arrangement of the polishing system. (**a**) Left side view of the system. (**b**) Right side view of the system.

**Figure 2 ijerph-17-06006-f002:**
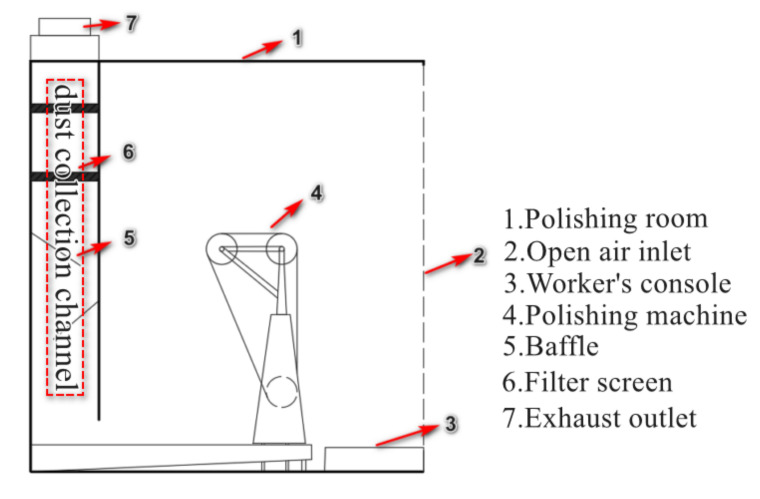
Plan view with existing polishing and dust collection system.

**Figure 3 ijerph-17-06006-f003:**
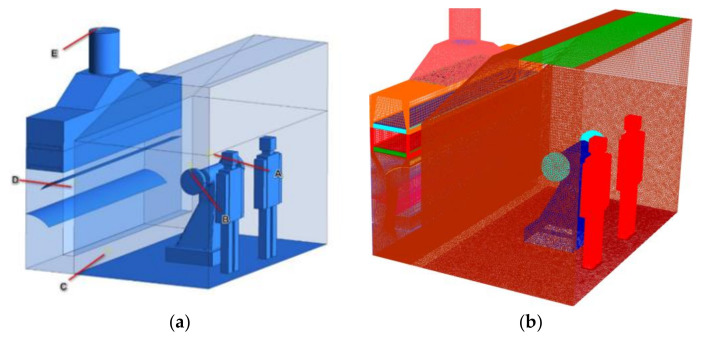
Three-dimensional view of the model adopted for integrated system. (**a**) Geometry and measurement points. (**b**) Computational hybrid grid.

**Figure 4 ijerph-17-06006-f004:**
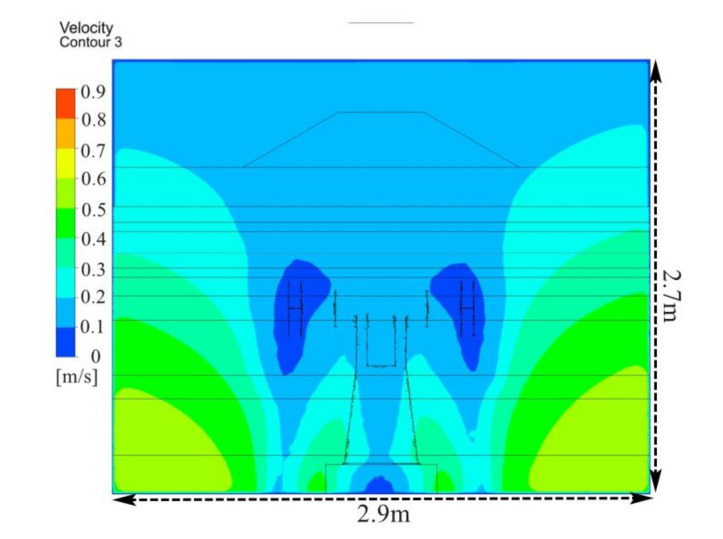
Velocity contour distribution in front of the polishing wheel with existing system.

**Figure 5 ijerph-17-06006-f005:**
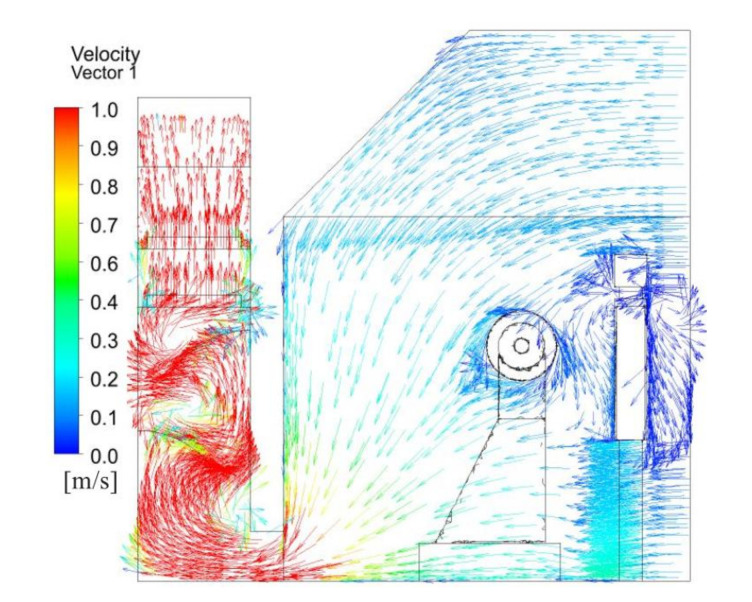
Velocity vector distribution across the polishing wheel center with existing system.

**Figure 6 ijerph-17-06006-f006:**
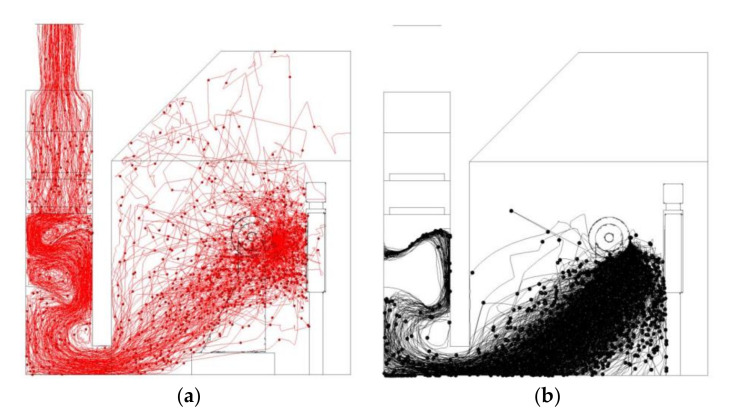
Particle trajectories colored by particle size in existing system. (**a**)Particle diameter smaller than 10 μm. (**b**) Particle diameter larger than 10 μm.

**Figure 7 ijerph-17-06006-f007:**
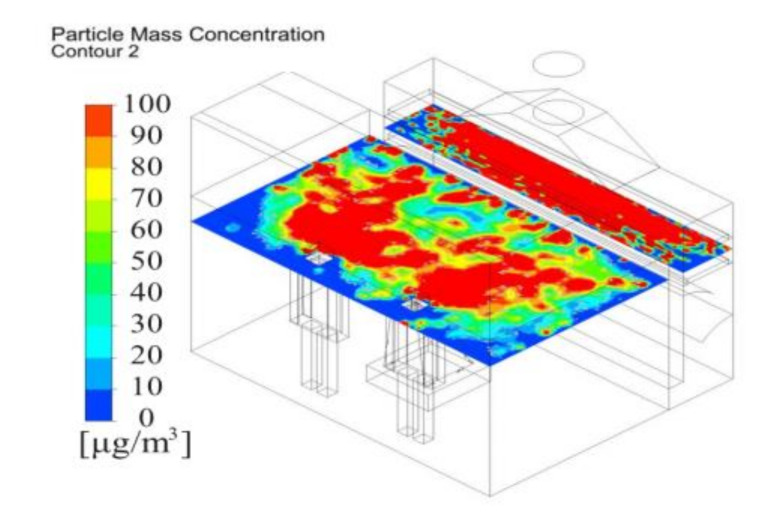
Dust concentration 1.5 m above floor with existing system.

**Figure 8 ijerph-17-06006-f008:**
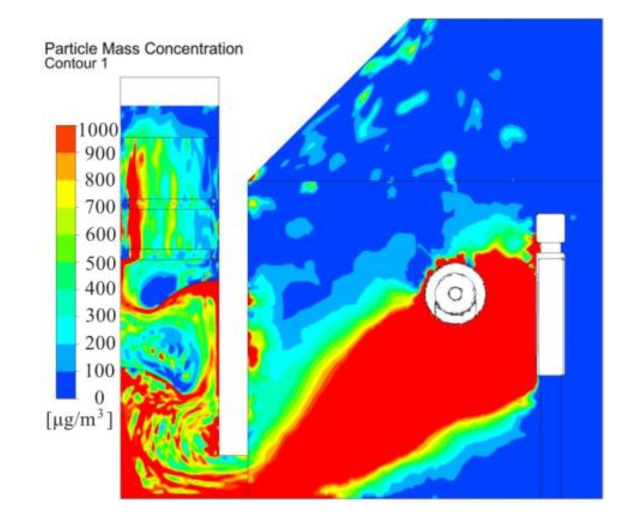
Dust concentration across the polishing wheel center with existing system.

**Figure 9 ijerph-17-06006-f009:**
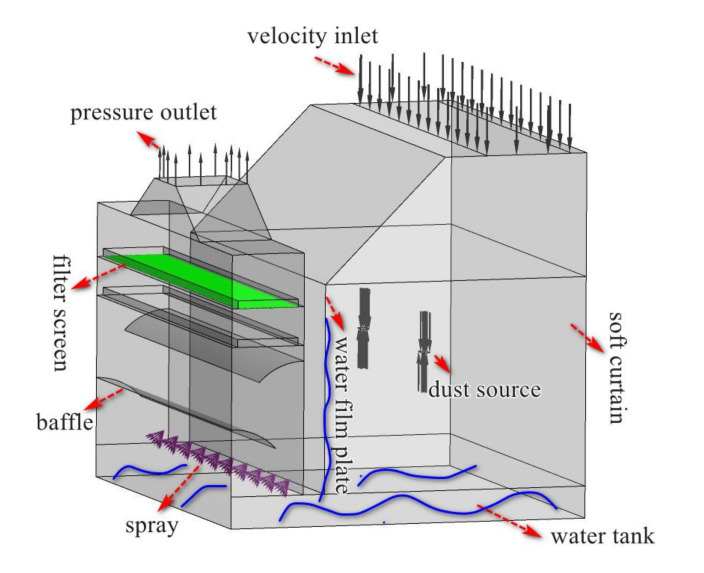
New ventilation arrangement and the design of an innovative wet dust removal system.

**Figure 10 ijerph-17-06006-f010:**
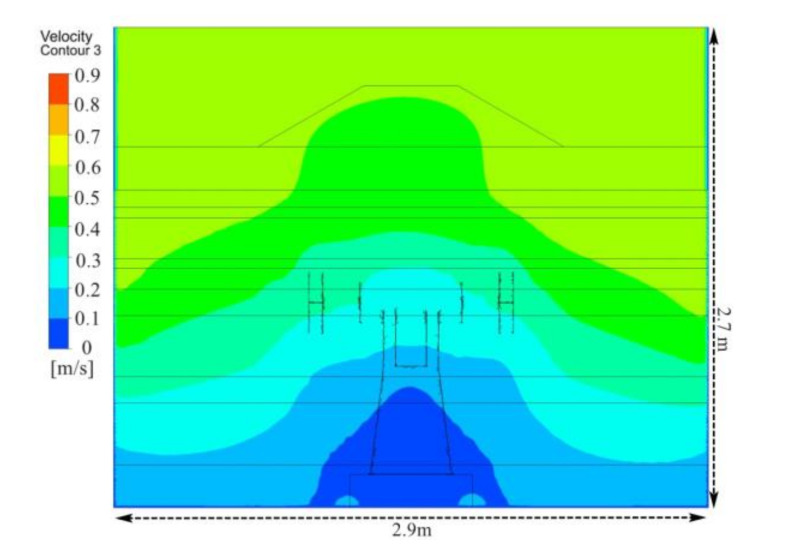
Velocity contour distribution in front of the polishing wheel with proposed system.

**Figure 11 ijerph-17-06006-f011:**
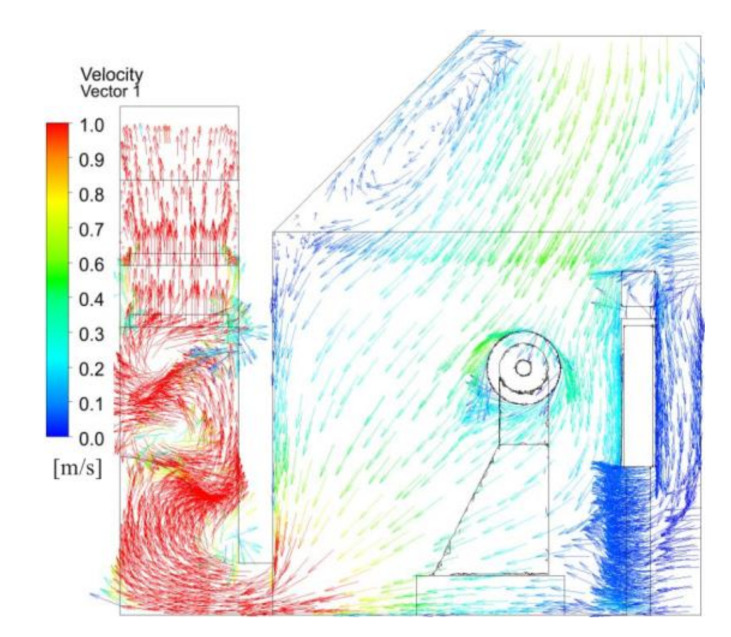
Velocity vector distribution across the polishing wheel center with proposed system.

**Figure 12 ijerph-17-06006-f012:**
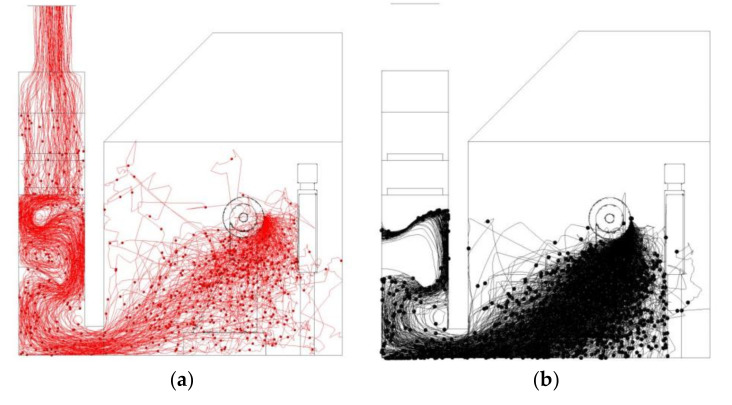
Particle trajectories are colored by particle size in proposed system. (**a**) Particle diameter smaller than 10 μm. (**b**) Particle diameter bigger than 10 μm.

**Figure 13 ijerph-17-06006-f013:**
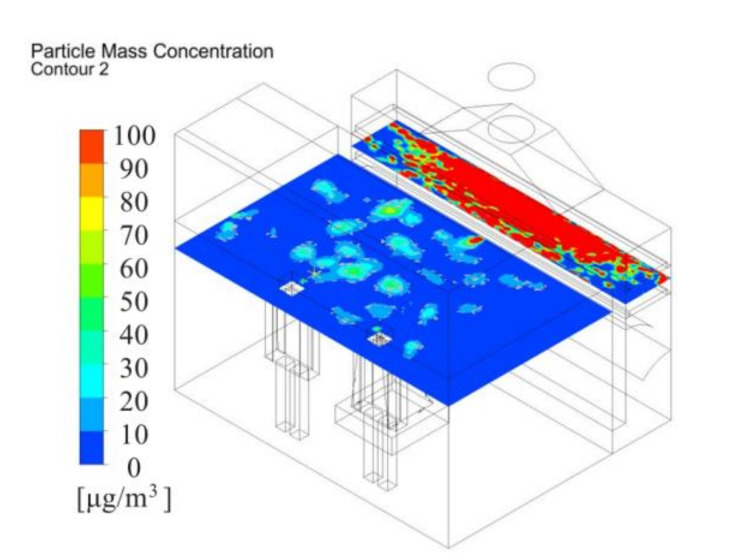
Dust concentration distribution 1.5 m above the floor under new push-pull ventilation.

**Figure 14 ijerph-17-06006-f014:**
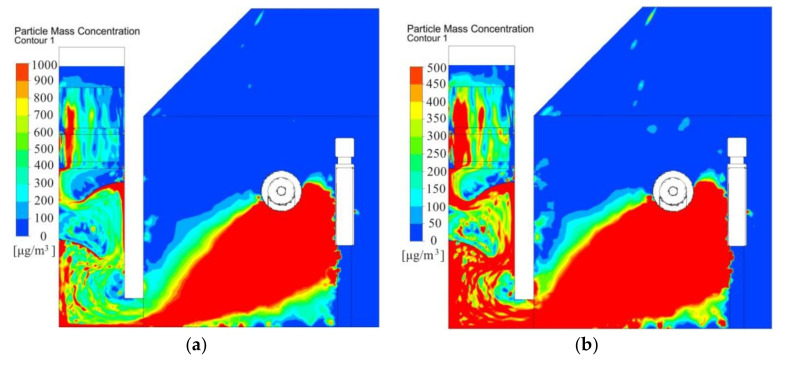
Dust concentration across the center of the polishing wheel. (**a**) Maximum dust concentration of 1000 μg/m3. (**b**). Maximum dust concentration of 500 μg/m3.

**Figure 15 ijerph-17-06006-f015:**
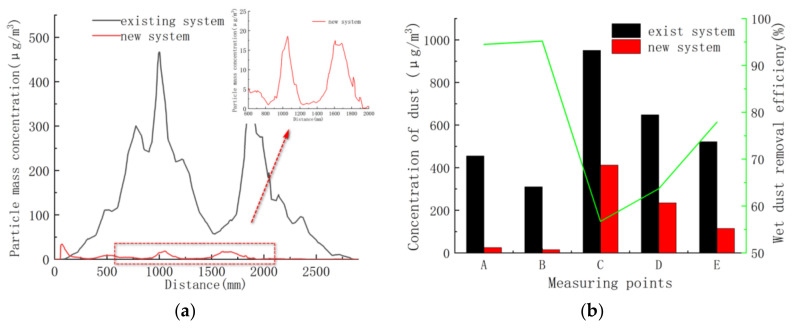
Evolution of dust concentration and dust removal rate before and after application of integrated dust control system. (**a**) Investigation line 1.5 m above the floor. (**b**) Measurement points in polishing chamber.

**Table 1 ijerph-17-06006-t001:** Ventilation measurements during polishing.

Location	Velocity (m/s)	Dust Concentration (μg/m^3^)
A	0.18	455
B	0.3	310
C	1.1	1520
D	4.0	
E	19.1	

**Table 2 ijerph-17-06006-t002:** Main geometric features used in modeling.

Name	Dimension (m)
Chamber width	2.3
Chamber length	2.9
Chamber height	2.7
Polishing machine width	0.7
Polishing machine height	1.2
Polishing machine length	1.3
Dust collection channel width	0.4
Dust collection channel length	2.9
Exhaust outlet diameter	0.3

**Table 3 ijerph-17-06006-t003:** Setting of parameters of dust source.

Main Parameters of Dust Source	Particle Size Distribution of Dust Particle Size Range/(m)	Rosin–Rammler 1 × 10^−6^–2.0 × 10^−5^
	Medium diameter/(m)	1.0 × 10^−5^
	Turbulent dispersion	Stochastic tracking
	Dust generation rate/(kg/s)	0.000002
	Wall condition	Reflect/trap/escape

**Table 4 ijerph-17-06006-t004:** Setting of boundary conditions and computational method.

Item	Name	Parameter
Boundary condition	Suction boundary type Relative pressure/(Pa)	PRESSURE_INLET 0
	Discrete phase model	ON
	Hydraulic diameter/(m)	2.6
	Turbulent intensity/(%)	4%
	Exhaust boundary type	Pressure outlet
Parameters of discrete term	Interaction with continuous phase Update DPM sources every flow iteration Maximum calculation step number Step length Drag law	Open Open 10000 0.01 Spherical
Calculation model	Solver Turbulence model	Discrete solver Standard k Two-equation
Solving parameter	Pressure–velocity coupling equation Discretization scheme	SIMPLEC First-order upwind scheme

**Table 5 ijerph-17-06006-t005:** Comparison between computation fluid dynamics (CFD) model and field-measured data.

Velocity Comparison	PM10 Dust Concentration Comparison
	Simulation (m/s)	Measured (m/s)	Error (%)	Simulation (μg/m^3^)	Measured (μg/m^3^)	Error (%)
A	0.16	0.18	11.1	470	455	3.1
B	0.28	0.3	6.6	320	310	3.1
C	1.2	1.1	8.3			
D	4.2	4	4.7			
E	20	19.1	5

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
