# Peer review of "CFD Modeling of Ventilation and Dust Flow Behavior in Polishing and the Design of an Innovative Wet Dust Removal System"

_ijerph, 2020, doi:10.3390/ijerph17166006_

Round 1

Reviewer 1 Report

I have completed the review of manuscript "CFD modeling of ventilation and dust flow behavior in polishing and the design of an innovative wet dust removal system" by Qian et al. This study provides computational fluid dynamics simulations for the understanding of the airflow patterns and dispersion of breathable dust in workplaces. The authors proposed a dust control system based on wet dust removal technology. Results are relevant since the efficiency in the reduction of respirable particles is around 95%. The manuscript contains interesting data and the application of the study is relevant to prevent accidents when managing aluminum dust. I recommend the publication of the manuscript and I have the following minor observations:

No information about the computational fluid dynamics simulations is provided in the abstract. In this section, it is mentioned the efficiency of the wet dust removal technology but such systems are commonly used in the industry. In particular, high tech air filtration systems are used in highly reactive dust such as aluminum. For example the hydro cyclone filtration systems, HEPA, etc. Please provide information about the relevance of the computational simulations and the improvement of the wet dust removal system when compared to existing technology.

Validation of model: Authors measured the airflow velocities and respirable dust concentrations during the ventilation survey for model validation. In figure 6, particle trajectories for two different particle sizes are shown, but it is unclear from the text how the particle trajectories were validation. Please provide more information about the measurements.

Author Response

Dear reviewer:

  We have studied the valuable comments from you, the assistant editor and reviewers carefully, and tried our best to revise the manuscript. The point to point responds to the reviewer’s comments are listed as following:

 Responds to the reviewer’s comment:

 Reviewer 1

 Comment 1 : Please provide information about the relevance of the computational simulations and the improvement of the wet dust removal system when compared to existing technology.

Response: Thank you for your comment. we have added the information of the wet dust removal system when compared to existing technology in abstract. Compared with air filtration systems, wet dust removal system could mitigate the occurrence of dust explosions. According to simulation results, spray system nozzles installed at the bottom of the dust collection channel, which formed a mist covered the dust sources and encouraged dust settling.

Comment 2 : In figure 6 , particle trajectories for two different particle sizes are shown , but it is unclear from the text how the particle trajectories were validation. Please provide more information about the measurements.

Response:According to the reviewer’s comment, we used DT-96 particle detector which used laser sensor to identify the fine particulate matter in the air to accurately measure the mass concentration of PM10 particles in real time. the mean value as the statistic of real-time multi-group measurement data included in Table 2 are presented. The measure results can explain the PM10 trajectory that dispersed around the polishing machine and dispersed into the dust collecting channel. Particle image analyzer is adopted to analyze the sizes of dust sample , which are collected from dust sources. The analysis indicates that a size smaller than 20 generated in polishing. Analysis of particles collected from the floor, Many particle larger than 10 microns deposited on the floor.

Reviewer 2 Report

The subject of the manuscript is interesting, but in order to publish this manuscript, the following comments should be taken into account and be clarified:

  1. What is the size of your boundary layer?
  2. How do you guarantee the solution is converged?
  3. What are the criteria of mesh independence study?
  4. The authors mentioned that the standard k-epsilon model was used to simulate the airflow, why k-epsilon?
  5. The dust size distribution mode is Rosin-Rammler. Why did you use this model? References are needed to cite to support this point.
  6. How do you measure the dust concentration /air velocity? What kind of equipment did you use? Please briefly describe how you got the data.
  7. In this study, a new wet dust collection system was used to improve the dust collection efficiency. With the wet dust collection system, most of the dust was captured by water droplets. Did the author simulate the collision between dust and water droplets? If so, how did you simulate this dynamic process? Please clarify this point.

Author Response

Dear reviewer:

  We have studied the valuable comments from you, the assistant editor and reviewers carefully, and tried our best to revise the manuscript. The point to point responds to the reviewer’s comments are listed as following:

Comment 1 : What is the size of your boundary layer?

Response: Thank you for your comment. The value of Y+ is about 110. The value is suitable for the turbulent model and standard wall function method.

Comment 2: How do you guarantee the solution is converged?

Response: In order to guarantee the solution is converged, we choose the available boundary conditions. The calculation scheme adopted first order Upwind. Appropriate to reduce Under-Relaxation factors. Steady computation was conducted using the SIMPLEC algorithm, during which the number of iterations was set as 2000. After reaching the convergence, the data was exported for further analysis. And the discrete model was added to carry out the calculation.

Comment 3: what are the criteria of mesh independence study?

Response: the Direct Differentiation, used

here,  has the advantage of being exact, due to direct differentiation of governing equations with

respect to design parameters, but limited in scope

Response: To avoid any mesh dependency, meshes with different densities, denoted as coarse mesh, middle mesh and fine mesh, were generated using ICEM-CFD ,whose mesh numbers were 1.2 million, 2.4 million and 3.6 million, respectively. A straight line at 1.5m above the floor was set in the model. Twenty points were arranged above the line at the same intervals, and airflow velocity and dust concentrations at these points in the three types of meshes were derived and compared. Comparison showed that 2.4 million elements could achieve independent solution. Therefore, middle mesh was selected in this study for further calculations. The final meshing result is shown in Figure.3

Comment 4: The standard k-epsilon model was used to simulate the airflow, why?

Response: The airflow in a polishing process is a typical turbulent flow. So we first choose stand k-epsilon model which was the most widely-used engineering turbulence model for industrial applications. The results may reasonably accurate with standard model.we should consider comparing the results using different turbulence models.

Comment 5: The dust size distribution mode is Rosin-Rammler. Why did you use this model? References are needed to cite to support this point.

Response: Particle image analyzer was adopted to analyze the sizes of dust sample, which were collected from dust sources. The analysis indicated that dust with a size smaller than 20μm generated in polishing. According to the regression analysis of least squared method, the sizes of fine dust followed the Rosin-Rammer distribution. During the polishing process, there was mainly due to occupational health hazards and safety hazards by PM10 suspended in polishing.Reference “ Study of hydrogen explosion control measures by using l-phenylalanine for aluminum wet dust removal systems.” can support this point.

Comment 6: How do you measure the dust concentration/air velocity? What kind of equipment did you use? Please briefly describe how you got the data?

Response: We used Kestrel handheld anemometer to measurement airflow. we used DT-96 particle detector which used laser sensor to identify the fine particulate matter in the air to accurately measure the mass concentration of PM10 particles in real time. We measured data for 10 second in real time. Through statistical analysis, table 1 shows the statistic average value.

Comment 7: In this study, a new wet dust collection system was used to improve the dust collection efficiency. With the wet dust collection system, most of the dust was captured by water droplets. Did the author simulate the collision between dust and water droplets? If so , how did you simulate this dynamic process? Please clarify this point.

Response: Thank you for your comment. In this paper, We did not simulate the collision between dust and water droplets. Experimental measurement of dust removal efficiency before and after spray.

Reviewer 3 Report

The manuscript deals with CFD modelling of ventilation and dust flow behaviour in polishing and the design of an innovative wet dust removal system. The scientific quality of the paper is very good and the topic is relevant to the International Journal of Environmental Research and Public Health. The manuscript is in general well written and it is important to highlight the difference and new insights in the present work. The results can be useful for management and control of high dust levels in the polishing processes/ potentially in other industries with similar production conditions. 

Author Response

Dear reviewer:

Thank you for your letter and for the referee’s comments concerning our manuscript entitled. We have studied your comments carefully and have made correction which we hope meet with approval.

Reviewer 4 Report

Interesting article about mitigation of  dust hazards in polishing chambers. The authors carried out measurements of air velocity and dust concentration in the chamber. They performed CFD calculations for the air velocity field, movement of dust particles and dust concentrations. Proposition of a new dust mitigation system was presented.

Some specific remarks in attachment.

Author Response

Dear reviewer:

  We have studied the valuable comments from you, the assistant editor and reviewers carefully, and tried our best to revise the manuscript. The point to point responds to the reviewer’s comments are listed as following:

Response:

  1. Lines 5-9

Please check affiliation. Authors identified by numbers 3 and 4 have no affiliation.

Response: we have check affiliation, identified by numbers 3 and 4 .

  1. Lines 40-42

Propose to update your information about new cases of pneumoconiosis

Response: we have update the information about new cases of pneumoconiosis according to data from the Chinese Center for Disease Control and Prevention.

  1. Lines 186-187

“The measured flow rate of some points indicated that about 0.83~1.0 m3/s of fresh air was provided to the open air inlet”

Please explain , how did you calculate airflow volume at open air inlet? What was the airflow volume at the exhaust outlet? Were the two mentioned values similar?

Response: The average velocity inlet times inlet cross-sectional area is equal to the inlet airflow volume. The airflow volume at the exhaust outlet is equal to the average velocity outlet times outlet cross-sectional area. There is an error between the inlet gas flow rate and the outlet gas flow rate, but the error is not more than 10%.

  1. Line 192

What model of Kestrel anemometer was used? Describe technical parameter of anemometer

Response: We used model of measure the average value. The machine NK3500 , CE certification.

5.Table 2

 Some model dimensions are missing: polishing machine location , machine length , exhaust outlet diameter , dust channel length , baffle location , Perhaps it is worth presenting a suitable figure?

Response: Thank you for your comment. We have add the model dimensions in paper.

6.Figure 4 and 10 Please , add dimensions on the axes. It will be helpful for a readers.

Response: Thank you for your comment. we add dimensions on the axes.

  1. Lines 405-406

“Otherwise , measures to repel dust must be taken with increasing ventilation to discharge and dilute the gas in worker operating areas.”

Answer: we changed gas into dust concentration ,

  1. Line 403 and figure 7

 The dust concentration was more than 300 μg/m3 at the worker’s breath level . in figure 7

Response: According to the set maximum dust concentration, we modified the dust concentration to 100 μg/m3 in the workers' breathing area.

9.Lines 410-412

“The dust concentration was the highest in the polishing chamber, where the dust concentration reached a level greater than 1000 μg/m3

Response: we changed the highest of dust concentration.

10.Lines 427-428

 The polishing chamber was connected to a suitable dust extraction fan, with a capacity of 1.2 m3/s” Please, add information about “ Old-before ventilation modification) fan”

 Response: Thank you for your comment. we choose a centrifugal fan, Flow rate is 2664 ~ 5268m3/h, The total pressure is 1578~1000Pa. old-before fan have not changed.

Reviewer 5 Report

In the paper CFD modeling was used to visualization of ventilation flow and dust distribution in the aluminum polishing system and propose the new wet dust collector for these system. For this purpose, additional field experiments were carried out based on measuring the concentration of dust in the air within the polishing machine before and after the application of the new dust removal technology.

The paper requires some corrections before its possible publication.

Main comments:

DT-96 dust sampler was used to measure air concentrations of dust in the area of the polishing system. The reliability of the measurements performed with this type of device raises some doubts. First of all, however, it is necessary to specify in section 2.2 what kind of dust (PM2.5, PM10, PM20 or maybe TSP?) was measured and at what averaging time the results of the measurements included in Table 2 are presented. DT-96 dust sampler probably allows the approximate measurements of PM2.5 and PM10 concentrations, and not TSP or PM20 (referred to as the "respirable" dust in the study).

The method of determining the dust generation rate, included in Table 3 and too generally described in section 3.2, also requires a more precise explanation.

Figures 6 and 12 show the results of modeling the distribution of particulates below and above 10 μm within the polishing system. The paper should explain all assumptions related to this modeling, including how the dust emission level was adopted for both of these dimensional fractions.

In the term "ventilation wind speed" used in line 67, the word "wind" seems to be inappropriate in this context.

Editorial comments:

Figures are not marked in accordance with the instructions for Authors provided on the website: https://www.mdpi.com/journal/ijerph/instructions (the full name Figure should be used instead of Fig.).

The legend in Figure 2 could be slightly shifted to the right so that it does not overlap with the description of the arrow no. 2.

The method of placing the units of various parameters in Table 3 (in brackets or after the symbol: /) should be standardized.

It is recommended to check the correctness of reference in the article to references [26] and [27], as well as the use of the reference (ANSYS, 2015) in lines 249-250.

The references list does not comply with the guidelines. The appropriate capital letters abbreviations should be used instead of full names (e.g. instead of “Applied surface science”, “RSC advances”, “Industrial Safety and Environmental Protection” etc. should be used respectively: Appl. Surf. Sci., RSC Adv., Ind. Saf. Environ. Prot. etc.). Non-uniform formatting of years and/or volume numbers (e.g. in [11], [20], [24] and [27]). Unnecessary tags "pp." or "p." before the page number range. The DOI numbers are also missing.

Author Response

Dear reviewer:

  We have studied the valuable comments from you, the assistant editor and reviewers carefully, and tried our best to revise the manuscript. The point to point responds to the reviewer’s comments are listed as following:

1.It is necessary to specify in section 2.2 what kind of dust (PM2.5, PM10,pm 20 or maybe TSP?) was measured and at what averaging time the results of the measurements included in Table 2 are presented.

Response: We are sorry for not addressing what kind of dust was measured clearly. In this study, we used DT-96 particle detector which used laser sensor to identify the fine particulate matter in the air to accurately measure the mass concentration of PM10 particles in real time. We measured data for 10 second in real time. Through statistical analysis, table 1 shows the statistic average value.

2.The method of determining the dust generation rate , included in Table 3 and too generally described in section 3.2 , also requires a more precise explanation.

Response: We used DT-96 measured the mass concentration of PM10 particles ,and the mass concentration of particles larger than 10 microns was calculated according to the normal distribution rule of particle size. So determine the dust generation rate included in Table 3.

3.Figures 6 and 12 show the results of modeling the distribution of particulates below and above 10 within the polishing system. The paper should explain all assumptions related to this modeling, including how the dust emission level was adopted for both of these dimensional fractions.

Response: Particle image analyzer was adopted to analyze the sizes of dust sample, which were collected from dust sources. The analysis indicated that dust with a size smaller than 20μm generated in polishing. According to the regression analysis of least squared method, the sizes of fine dust followed the Rosin-Rammer distribution. The dust emission level of Aluminum dust was 3 mg/ m3 in China.The dust emission level of PM10 was 150 μg/m3 in China.

4.In the term “ ventilation wind speed” used in line 67 , the word “ wind” seems to be inappropriate in this context.

Response: Thank you for your comment. we have change “ ventilation wind speed” to “ airflow velocity” in paper.

5.Thank you for your comment. We modified the format of the figure according to the requirements of the magazine. We have change the legend in Figure 2. Change the units in the same parameters in Table 3. We have check the correctness of reference in the article to reference [26][27] as well as the use of the reference (ANSYS2015) in lines 249-250. The appropriate capital letters abbreviations be used instead of full names.